# Immunogenetic Landscape in Pediatric Common Variable Immunodeficiency

**DOI:** 10.3390/ijms25189999

**Published:** 2024-09-17

**Authors:** Aleksandra Szczawińska-Popłonyk, Wiktoria Ciesielska, Marta Konarczak, Jakub Opanowski, Aleksandra Orska, Julia Wróblewska, Aleksandra Szczepankiewicz

**Affiliations:** 1Department of Pediatric Pneumonology, Allergy and Clinical Immunology, Institute of Pediatrics, Poznan University of Medical Sciences, Szpitalna 27/33, 60-572 Poznań, Poland; alszczep@ump.edu.pl; 2Student Scientific Society, Poznan University of Medical Sciences, 60-572 Poznań, Poland

**Keywords:** antibody deficiency, common variable immunodeficiency, epigenetics, immune dysregulation, inborn error of immunity, whole exome sequencing

## Abstract

Common variable immunodeficiency (CVID) is the most common symptomatic antibody deficiency, characterized by heterogeneous genetic, immunological, and clinical phenotypes. It is no longer conceived as a sole disease but as an umbrella diagnosis comprising a spectrum of clinical conditions, with defects in antibody biosynthesis as their common denominator and complex pathways determining B and T cell developmental impairments due to genetic defects of many receptors and ligands, activating and co-stimulatory molecules, and intracellular signaling molecules. Consequently, these genetic variants may affect crucial immunological processes of antigen presentation, antibody class switch recombination, antibody affinity maturation, and somatic hypermutation. While infections are the most common features of pediatric CVID, variants in genes linked to antibody production defects play a role in pathomechanisms of immune dysregulation with autoimmunity, allergy, and lymphoproliferation reflecting the diversity of the immunogenetic underpinnings of CVID. Herein, we have reviewed the aspects of genetics in CVID, including the monogenic, digenic, and polygenic models of inheritance exemplified by a spectrum of genes relevant to CVID pathophysiology. We have also briefly discussed the epigenetic mechanisms associated with micro RNA, DNA methylation, chromatin reorganization, and histone protein modification processes as background for CVID development.

## 1. Introduction

Advancement in the field of immunogenetics in pediatric inborn errors of immunity (IEI) has a multidimensional character, as it not only plays a crucial role in the progress of understanding the immunological pathways involved in the human immune response but also assists therapeutic decisions and contributes to the progress of patient-tailored precision medicine.

Whereas primary antibody deficiencies represent the most common category of IEI, in young children, significant immaturity of the immune system and developmental delay of antibody production may impede an accurate definitive diagnosis. The greatest and the most dynamic maturational changes among the B cell subsets occur in children below the age of four years. Therefore, the reliable diagnosis of primary antibody deficiency in children less than four years old cannot yet be established, despite the fact that their clinical and immunological phenotypes share common features with specific disease entities. This group of young pediatric patients with deficiencies of antibody production requires a discerning clinical observation and immunodiagnostics. At this point, it also needs to be highlighted that the classifications used in adults are not helpful, and their application to children with antibody production defects is misleading, as they require separate diagnostic and prognostic criteria [1,2].

Pediatric common variable immunodeficiency (CVID) is a primary symptomatic hypogammaglobulinemia, a common disorder among IEI with important morbidity and individual and societal burden, but it is also a phenotypically and genetically variable disease, thereby challenging for pediatricians. Increased awareness of CVID is the first successful step in initiating the diagnostic process. For pediatric immunologists, the major task is to evaluate the parameters of the immune system against the maturation of the adaptive immune responses and to perform further advanced diagnostics including flow cytometric lymphocyte analysis and the genomic approach [3]. CVID is characterized clinically by a broad spectrum of severe associated outcomes, such as recurrent infections, organ-specific immunopathology, immune dysregulation, and malignancy [4]. According to the International Consensus (ICON) statement [5], the revised European Society for Immunodeficiencies (ESID) diagnostic criteria [6], and the current classification of the International Union of Immunological Societies (IUIS) Expert Committee [7], CVID is characterized by low serum IgG levels, accompanied by decreased IgM and/or IgA, impaired specific antibody response to vaccines, and exclusion of other specific causes of hypogammaglobulinemia [8]. Furthermore, the ESID definition of CVID includes clinical criteria, such as increased susceptibility to infection, autoimmune manifestations, granulomatous disease, polyclonal lymphoproliferation, and a positive family history of antibody deficiency. Immunodiagnostic criteria for CVID in children include antibody deficiency interpreted in relation to age-matched reference values, and low switched memory B cell numbers, below 70% of age-related normal values. Importantly, evidence of profound T cell deficiency, low CD4^+^ T cell counts, low relative numbers of CD4^+^ T cells in relation to the child’s age, and absent T cell proliferation are exclusion criteria for CVID [6]. In patients with CVID, a spectrum of phenotypic and functional abnormalities in the adaptive and innate immunity reflect impaired immune homeostasis, including defective B-cell differentiation and maturation, isotype-switched memory B-cell development [9], T-cell dependent costimulation, regulatory T cells [10], and disturbances in various components of innate immunity [11]. The data on complex immunophenotypes in CVID have led to further stratification of the disease and developing Freiburg [12] and EUROclass [13] classifications. Nonetheless, the application of the criteria based on terminal differentiation of B cells and formation of isotype-switched memory B cells used in these classifications in pediatric patients is a matter of discussion because of active maturational processes and dynamic shifts within B and T cell subsets in childhood [14]. Due to this marked heterogeneity of B and T cell abnormalities in CVID, it has become clear that CVID is an umbrella diagnosis encompassing a spectrum of disorders with a defective biosynthesis of antibodies as a common denominator. 

These complex processes of B cell antigen signaling, activation, survival, migration, and maturation to generate terminal stages of switched memory B cells and plasma cells mirror a composite genetic etiology of CVID. The B cell developmental impairment and hypogammaglobulinemia may result from genetic defects of many receptors and ligands, activating co-stimulatory molecules, and intracellular signaling molecules [15]. However, despite a relatively high prevalence of CVID among IEI in children, the rate of molecular genetic diagnosis remains low, with pathogenic gene variants identifiable in a limited proportion of patients, ranging from merely 2–10% [16] up to 54% in populations with a high rate of consanguinity [17]. Therefore, it strongly implies that beyond the monogenic model of inheritance, another explanation of CVID origin is multifactorial, digenic, or polygenic, and alternatively, that accumulation of rare functional variants, somatic variants, or epigenetic phenomena [17,18,19,20] may show a causal relationship with the regulation of B cell development and functions.

This review aimed to resume and conclude the immunological and genetic underpinnings of pediatric common variable immunodeficiency. It was also conducted to provide data facilitating a better understanding of heterogeneous cellular and genetic immunophenotypes in the context of mechanisms determining infectious and non-infectious manifestations of the disease.

## 2. Monogenetic Model of CVID in Children

The molecular genetic background associated with monogenic causes has been hitherto identified in less than 20% of patients affected with CVID, usually in its familial forms which constitute only a small fraction of cases. Furthermore, some of the mutations are characterized by incomplete penetrance, and also sporadic cases remain genetically unexplained, suggesting a complex, non-Mendelian pattern of inheritance [17,18,19,20]. Better understanding of disease immunopathogenesis and the genotype-phenotype relationship could help stratify patients into clinical disease entities to predict complications and possibly individualize treatment. While clinical presentation, disease severity, and immunophenotype in pediatric CVID are highly variable with infectious and non-infectious complications resulting from immune dysregulation [21,22,23,24,25], CVID is currently perceived as a group of disorders, with antibody deficiency as their common denominator and a cardinal feature, covering a spectrum of genetic subtypes. Although the main tool for diagnosis of CVID remains clinical, it is highly recommended to obtain a genetic signature and molecular analysis in all affected subjects with unclear and severe clinical phenotypes [26].

The genetic etiology of CVID underpins complex processes of B cell antigen signaling, activation, survival, migration, and maturation to generate terminal stages of switched memory B cells and plasma cells. The B cell developmental impairment and antibody deficiency may result from genetic defects of many receptors and ligands, activating and co-stimulatory molecules, and intracellular signaling molecules. Consequently, these genetic variants may affect crucial immunological processes of antigen presentation, antibody class switch recombination (CSR), antibody affinity maturation, and somatic hypermutation (SHM). Furthermore, variants in genes linked to antibody production defects and immune dysregulation with autoimmunity, lymphoproliferation, enteropathy, splenomegaly, and granulomatosis have been identified thus far in a proportion of affected patients [23,27]. Genes that have been identified in monogenic CVID on the European background include *ICOS* (inducible T cell co-stimulator), *TNFRSF13B* (transmembrane activator and calcium modulator and cyclophilin ligand interactor, TACI), *TNFRSF13C* (B cell-activating factor belonging to the tumor necrosis factor (TNF) family, BAFF-receptor, BAFF-R), *TNFSF12* (TNF-like weak inducer of apoptosis, TWEAK), *CD19*, *CD81*, *CR2* (CD21), *MS4A1* (*membrane-spanning 4A1*, CD20), *TNFRSF7* (CD27), *IL21*, *IL21R*, *LRBA* (lipopolysaccharide (LPS)-responsive beige-like anchor protein), *CTLA4* (cytotoxic T lymphocyte-associated antigen 4), *PRKCD* (Protein kinase C delta), *PLCG2* (phospholipase C gamma 2), *NFKB1* (nuclear factor kappa B1), *NFKB2* (nuclear factor kappa B2), *PIK3CD* (Phosphoinositide 3-kinase (PI3K) catalytic subunit delta), *PIK3R1* (Phosphoinositide 3-kinase (PI3K) regulatory subunit 1), *VAV1* (Vav guanine nucleotide exchange factor 1), *RAC2* (Ras-related C3 botulinum toxin substrate 2), *BLK* (B-lymphoid tyrosine kinase), *IKZF1* (IKAROS), and *IRF2BP2* (Interferon regulatory factor 2 binding protein 2) [15]. An in-depth retrospective review of genes reported in monogenic CVID patients categorized variants involved in multiple molecular pathways and immune compartments such as, but not limited to, B cell receptor (BCR) costimulatory B cell surface proteins, tumor necrosis factor superfamily receptors and ligands, lipid signaling molecules, actin cytoskeleton regulators, transcription factors mediating differentiation and crosstalk, metabolic processes of glycosylation, and mitochondrial pathways [28]. The spectrum of most frequently reported genetic variants underpinning monogenic CVID stratified according to diverse pathomechanisms is displayed in Table 1. 

Beyond infections, pediatric CVID is characterized by impaired immune homeostasis resulting in a high rate of autoimmune complications and inflammatory disorders associated with the above-mentioned genetic variants, such as PRKCD and LRBA deficiencies, CTLA-4 haploinsufficiency, as well as the activated PI3Kδ syndrome [29]. These monogenic forms of CVID are connected with two predominating immunopathological pathways, disrupted peripheral tolerance due to the impaired T regulatory cell functions in the first ones [30,31] and hyperactivation of T and B cells in the latter [32]. 

Several novel genetic defects, such as variants in *STAT1*, *DOCK8* (Dedicator of cytokinesis 8), *AIRE* (Autoimmune regulator) [23,33], *FOXP3* (Forkhead box protein 3) [23], *CXCR4* (CXC chemokine receptor 4) [16], *BACH2* (BTB and CNC homolog 2) [34], or *STAT3* (Signal transducer and activator of transcription 3) [23,35] causing immune dysregulation syndromes presenting as CVID have also been identified. 

Noticeably, variants in *DNMT3B* (DNA methyltransferase 3 beta), known as the disease-causing gene for Immunodeficiency-Centromeric Instability-Facial anomalies (ICF) syndrome, have also been demonstrated in CVID patients [16,23], thereby linking monogenic underpinnings with epigenetic mechanisms in the pathophysiology of CVID. Several monogenic defects have also been hypothesized to interact with the epigenome and impact epigenetic enzymes and their interactors. Therefore, epigenetic alterations and pathogenic variants in CVID-associated genes do not exclude each other as disease-causing immunopathology but provide a novel insight into the mutual relationships between genome and epigenome in CVID [36]. These phenomena of an interplay between genetic variants in transcription factors relevant to the pathogenesis of CVID in conjunction with epigenetic deregulation are well illustrated by NF-κB and JAK (Janus kinase)—STAT signaling pathways. For example, variants in the *NFKB1* gene in the form of haploinsufficiency have been reported as the most prominent disease-causing monogenic background for CVID among Europeans [37]. Notably, variants in *NFKB1* can impact epigenetic mechanisms such as chromatin remodeling of target genes and histone modifications by interactions with epigenetic enzymes, e.g., histone deacetylases and methyltransferases, thereby resulting in modulation of lymphocyte activity [36,38]. Variants in the JAK-STAT signaling pathway, in particular *STAT1* gain-of-function (GOF), and *STAT3* GOF that impair binding of STST molecules to their target genes are connected to epigenetic deregulation of histone methylation [36,39,40].

The multiplicity of genetic variants found in patients with CVID phenotypes, associated with a constellation of clinical symptoms, from infections to lymphoproliferation, point to the complexity of numerous pathways involved in B lymph cell development and antibody-dependent immune response. The ever-increasing advances in genomics and the consequent growing rate of monogenic causes of CVID indicate the need to reevaluate patients and their reassignment to specific immunogenetic categories of IEI. 

## 3. Digenic and Polygenic CVID

For genetically heterogeneous diseases, such as CVID, for which many disease-causing variants in many genes are known, digenic combinations are likely and they may display different modes of inheritance. Noticeably, in sporadic CVID cases, for which the genetic underpinnings remain unknown, the digenic etiology should be considered [41].

Beyond the genotype-age-related immunophenotype in pediatric CVID, identifying variants not following the Mendelian mode of inheritance in familial cases with variable degrees of penetrance and expressivity determining remarkable diagnostic challenges, the complexity of CVID is even more conveyed by a predisposition to the disease. Noticeably, variants in *TACI*, a member of the TNF superfamily, may not be causative for CVID but may coexist and interact synergistically with other variants showing deleterious effects. Thereby, the epistatic interactions, synergistic interplay of two genetic loci that substantially modify disease severity or result in entirely new phenotypes as well as gene additivity effects, may determine the disease symptomatology [32,42,43]. The epistasis phenomena may be exemplified by digenic variants in genes in which products are playing roles in the same physiological pathways, e.g., variants in *TACI*, stimulating a T-cell independent class switch recombination, and *TCF3* (Transcription Factor 3) aka *E2A*, a central point of T-cell independent and T-cell dependent immunoglobulin class switching and secretion with a clinical phenotype of immunodeficiency and autoimmunity showing a digenic nature of CVID [44]. The digenic etiologies associated with CVID phenotypes with antibody production deficiencies and autoimmunity relating to clinical epistasis, and including variants in genes such as *NFKB1* and *NOD2* (Nucleotide-binding oligomerization domain containing 2) [45], *LRBA* and *NEIL3* (Nei-Like DNA Glycosylase 3) [46], and *CTLA4* and *JAK3* (Janus Kinase 3) [47], have been summarized in Table 2.

Whereas the multiple variants found in the monogenic CVID are limited to familial cases with autosomal dominant inheritance and different expression and penetrance, a complex polygenic model of inheritance is likely to be the underlying mechanism for the majority of sporadic cases [48,49]. The genetic analysis of data may also be hampered by a marked intersubject and intrafamilial clinical phenotypic heterogeneity pointing to the modifying effects of additional genetic variants to the broad symptomatology in CVID. Furthermore, a spectrum of environmental contributors, such as dysbiosis of gut microbiota, shifts of microbial composition, and reduced diversity leading to gut-leakage syndrome, the use of drugs reducing B-cell development, such as rituximab or antiepileptic drugs, valproic acid, as well as viral infection inducing dysregulation of immune cells, are risk factors for severe course of CVID [50]. A polygenic burden of CVID has been supported by clinical follow-up and genetic analysis in monozygotic twins concordant for CVID in whom pathogenic variants in genes reported in the monogenic model of the disease have not been found. Nevertheless, as many as seven non-synonymous coding variants in genes involved in relevant immunological pathways, such as *JUN* (Jun Proto-Oncogene, AP-1 Transcription Factor Subunit), *PTPRC* (Protein Tyrosine Phosphatase Receptor Type C), *TLR1* (Toll-like Receptor 1), *ICAM1* (Intercellular Adhesion Molecule 1, CD54), and *JAK3*, with predicted deleterious effect and possible clinical impact, have been demonstrated [51].

At this point, referring to the global distribution and the worldwide country-wise prevalence of CVID, a correlation between the rate of CVID diagnosis and medical progress and socio-economic developmental status, expressed as the Human Development Index (HDI), has been observed [52]. This conclusion is drawn from assuming that the CVID prevalence does not truly differ between countries and that documentation in registries and referring to databases linked to the country’s developmental status is the major factor contributing to the obstacles in the estimation of the CVID epidemiology. However, an alternative explanation relies on admitting that CVID is presumably a polygenic disease and regionally distinct prevalence of the disease may be explained by influences from genetically diverse founders [53].

## 4. Somatic Variants in CVID Immunogenetics

In contrast to prezygotic germline variants present in the genome of all the body cells, including germ cells, somatic variants arise in the postzygotic DNA from embryonic development through postnatal life to adulthood. The postzygotic variants are associated with the phenomenon of gene mosaicism and, consequently, tissues represent a landscape of cells with different genomes. Postzygotic variants are differently distributed in tissues resulting from somatic, gonadal, or gonosomal mosaicism, with the complex and heterogeneous phenotypes and the two latter disorders posing a moderate to high risk of transmission of the variant to the offspring and various implications for disease phenotypes [54]. Somatic variants are often neutral or disadvantageous to cell growth and survival. They may result in clonal expansion conferring a fitness advantage to the cells or clonal regression when expanding clones shrink in size as a response to environmental changes. Given that age is a key accelerator of somatic mosaicism, it may therefore be assumed that children show lower risks of disadvantageous consequences of somatic variants, such as clonal hematopoiesis. It has also been hypothesized that gene mosaicism is a relevant and hitherto underrecognized mechanism underlying inborn errors of immunity. Indeed, postzygotic variants in several genes involved in CVID, such as *PIK3CD*, *STAT3*, or *BTK*, have been detected in affected families [55], suggesting somatic or gonosomal mosaicism and thereby explaining an intrafamilial phenotypic diversity as well as occurrence of IEI symptomatology in sporadic cases without family history [56,57]. Somatic mosaicism was first proposed as a cause of autoimmune lymphoproliferative syndrome (ALPS), a disease characterized by lymphocyte dysregulation and autoreactivity, showing common autoimmune and lymphoproliferative features with CVID. Somatic mosaicism has also been demonstrated in patients with autoinflammatory disorders and variants in *NOD2*, aka *CARD15* (Caspase-recruitment domain family, member 2), *JAK1* (Janus kinase 1), *NLRC4* (NOD-like receptor C) [57], and *IL6ST* (Interleukin 6 signal transducer) [58] as well as *KRAS*, the gene involved in the RAF/MAPK pathway [59]. Whereas in CVID, T cell abnormalities, such as deficiencies in naïve T CD4^+^ or regulatory T cells, and expansion of T CD8^+^ cells have been observed, somatic variants in *STAT5B* and *TET2* might also be linked to the disease pathophysiology and increased susceptibility to autoimmune diseases, lymphoproliferative disorders [59,60], and malignant transformation [61]. In a patient with a history of opportunistic infections with B and NK cell deficiency, a somatic pathogenic variant in the GATA2 gene has been detected [60].

## 5. Epigenetic Etiology of CVID

Despite the substantial clinical and genetic heterogeneity of CVID and ever-expanding progress in genomics, a monogenic molecular diagnosis has been found in a small proportion of affected individuals. Furthermore, as most patients with a diagnosis of CVID do not follow a classical Mendelian pattern of inheritance, often representing single sporadic cases, it could suggest that behind the monogenic or polygenic background, epigenetic phenomena may show a causal relationship with the regulation of B cell development and functions [62,63,64]. Epigenetic mechanisms are integrated, dynamic, and potentially reversible changes in gene expression without altering the germline DNA gene sequences, thereby accounting for the alterations in cellular development and differentiation. They comprise the regulation of DNA methylation, histone protein modifications, chromatin remodeling, and changes in transcription [65]. These observations could guide further investigations and epigenetics may, therefore, contribute to explaining the immunopathogenesis of CVID in patients who lack a molecular genetic diagnosis.

Non-coding RNA molecules (ncRNAs) are transcribed from DNA and not translated into proteins, but exert regulatory effects on gene expression and protein translation by influencing DNA transcription and mRNA post-transcriptional changes. In particular, micro RNAs (miRNAs) have critical regulatory functions in cell proliferation, programmed cell death, organ development, and differentiation. MiRNAs are important elements of the molecular pathways in hematopoietic cells and dynamic regulation of their expression suggests their roles in the early cell differentiation and lineage definition [66]. Among miRNAs that showed expression in the immature B cell phase, miRNA-181 and miRNA-150 contributed to the preferential expansion of the B lymph cell compartment [67,68]. It has also been hypothesized that the CD40 B cell receptor signaling in germinal centers for the differentiation of naïve to memory B cells modulates levels of several miRNAs, such as miR-150-5p, miR17-5p, miR146a-5p, miR26a-5p or miR-292-5p, and their cognate targets. Consequently, the CD40-miRNA axis controls prospective cell fate during B cell development [69,70]. A negative modulatory effect on B cell differentiation might, in turn, arise from the expression of miRNA-34a [71], potentially exerting lymphopenia and B cell developmental disturbances characteristic of CVID. Moreover, miRNA-mediated epigenetic regulation of T cell development and function might play a role in the pathogenesis of CVID. The candidate could be the miRNA-17-92 cluster which has been involved in the development and function of follicular CXCR5^+^ CD3^+^CD4^+^CD45RO^+^CD185^+^ T helper cells, supporting B cells in germinal centers and facilitating class-switch recombination, somatic hypermutation, and antibody affinity maturation [72]. Overexpression of miRNA-210 in T cells has also been postulated in pediatric-onset CVID patients with unsolved PID genetic defects, thereby supporting the hypothesis of the underlying epigenetic pathogenesis [73]. In these patients, the upregulation of miRNA-210 correlated with a reduced count of T CD4^+^ helper cells and T regulatory cells and an increased count of T CD8^+^ cytotoxic cells.

B cell-specific miRNAs, such as miRNA-15a-5p, miRNA-199a-3p, and miRNA-103a-3p have been shown to target genes modulating antigen-specific antibody titers following vaccination [74]. This impact on vaccine-induced antibody generation is exerted by different Fc receptors, the phosphatidylinositol-mediated signaling pathways, growth factors signaling pathways, and other processes with relevance to adaptive immunity [75]. The inability to mount a sufficient antigen-specific humoral immune response to vaccines is a fundamental feature and an essential diagnostic criterion of CVID. 

In CVID, impaired immunosurveillance also contributes to the autoinflammatory phenotype, which may be determined by a regulatory activity of miRNAs on pro- and anti-inflammatory cytokine profile, production of inflammasome components, effector T lymph cells differentiation, and ultimately, organ-specific inflammatory disorders. Several miRNAs are potential contributors to the inflammatory events, e.g., upregulated miRNA-21 and downregulated miRNA-23b may exert their inflammatory effect by influencing the activity of NFκB [76]. Likewise, dynamic regulations of the miRNA-155 targetome result in the overrepresentation of target genes involved in the immune response, including, among others, a complex of proinflammatory cytokines IL-6, IL-8, ICAM (intercellular adhesion molecule), and STAT3 [77]. It may, therefore, be assumed that in chronic structural and interstitial lung disease, exacerbated by respiratory viral infections, epigenetic regulation may play a pivotal role. This hypothesis may be supported by the findings showing differently modulated expression of miRNA-6742, miRNA-1825, miRNA-4769, miRNA-1228, and miRNA-1972, which are involved in adaptive immune response, by the first immunoglobulin transfusion [78]. 

Further studies are required to better define mutual relationships between epigenetic alterations and expression of miRNAs and the spectrum of clinical infectious, autoimmune, autoinflammatory, and lymphoproliferative phenotypes in pediatric CVID [21,23,24,79,80,81]. Much light needs to be shed on the role of epigenetics in immune dysregulation in CVID, manifesting as granulomatous lymphocytic interstitial lung disease (GLILD), bronchiectasis, lymphadenopathy, hepatosplenomegaly, arthritis, or inflammatory bowel disease (IBD). Given that miRNAs are involved in the regulatory activity of a spectrum of B-cell and T-cell intrinsic genes, interrelations between the different miRNAs expression and alterations in the immunophenotype of the B and T cell compartment need to be better understood [82]. 

With the ever-increasing progress of advanced methods in immunogenetics, analysis of DNA methylation and its role in the control of gene expression thereby shaping the epigenome and related biological processes may become a potential diagnostic and prognostic marker in CVID. DNA methylation is an epigenetic mechanism in which methyl groups are covalently bound to cytosine-producing 5-methylcytosine in the CpG dinucleotide domains. The importance of DNA methylation is multidimensional and more complex than repression or activation of gene expression, as it also determines interactions between methylated DNA and proteins, including transcription factors, readers of DNA methylation, and effectors involved in methylation signal translation to biological processes [83]. In CVID, aberrant DNA methylation has been demonstrated in 30% of CpG domains during the B lymph cell differentiation process, at the transition from naïve to memory stages. In the early phase of B cell differentiation, activation of key transcription factors was associated with demethylation of transcription enhancers of genes crucial in the biology of B cells. In the late phase of differentiation, demethylation of heterochromatin was associated with methylation of repression of genes functionally essential for the biology of B cells [84,85,86]. In CVID, switched memory B cells show marked heterogeneity in DNA methylation. Skewed DNA methylation is associated with active demethylation of transcription factors engaged in differentiation and activation of B cells, such as NF-κB, bZIP family (BATF, JUNB, Fosl2, Fra2), CTCF, IRF, and PU.1, consequently leading to disturbed binding to DNA. Furthermore, abnormally increased or decreased regulation of genes *CD70*, *CD40*, *NFKB2*, *ICAM1*, and *CCL17* has been identified in activated memory B cells, close to hypermethylated DNA regions [87,88].

Consequently, the transcription program in activated B cells in CVID is limited, and epigenetic alterations contribute to impaired transcription during the B cell immune response. The epigenetic imprinting in the switched memory B cell population occurs in genes and regulatory sequences activated in B cells germinal centers and in plasma cells. Furthermore, immunophenotypic dysregulation within the B-cell compartment may contribute to abnormal interactions between B cells and other immunological cells in CVID.

Epigenetic mechanisms have been schematically displayed in Figure 1.

## 6. Expected Advantages of Genetic and Epigenetic Studies in CVID

In children with the CVID phenotype, the expected advantages of identifying the causal pathogenic variants primarily include establishing the definitive diagnosis of CVID in its monogenic form. It might be helpful in distinguishing between transient hypogammaglobulinemia of infancy (THI) and CVID in young children with overlapping clinical and immunological phenotypes. In children showing variable serum IgG levels over time, thereby inconsistent with the ESID diagnostic criteria, and in whom timely monitoring of IgG production is not helpful for a clinical diagnosis, demonstration of the genetic underpinning could guide further immunodiagnostics and therapy. Furthermore, in these children, in whom the identified variant is likely to cause a specific disease, such as *NFKB2* [89,90] or *CTLA-4* [91,92], establishing the diagnosis of a CVID-like disorder may lead to excluding them from the umbrella diagnosis of CVID. In monogenic CVID, the identified disease-causing variant may assist in defining the prognosis regarding the clinical course of the disease, possible specific complications, such as an increased risk of autoimmunity, autoinflammation, granulomatous disease, lymphoproliferation, and malignancy. Optimizing the directed, individualized treatment, including an immunoglobulin replacement therapy, antimicrobial therapy, immunosuppression, and eventually a hematopoietic stem cell transplantation (HSCT), is primarily a variant-specific therapeutic approach. HSCT should be considered as a therapeutic option in children presenting with immunodysregulatory disorders due to genetic variants in *PRKCD*, *DOCK8*, and *FOXP3* [93]. The presence of a known variant has implications for family members as it is of paramount importance for family counseling and may also facilitate early diagnosis and a prompt starting of replacement Ig therapy in affected family members when they develop symptoms. Finally, the therapeutic aspect means identifying patients with monogenic CVID who may benefit from potential gene therapy in the future.

For researchers, the advantages of identifying the causative variant are also expected, offering a prototypic model of the monogenic CVID and the relationship between genotype and phenotype. It is promising in expanding the knowledge about the function of the immune system and the immunopathogenesis of CVID, CVID-like disorders, and other hypogammaglobulinemias in children that also implicate correlations between the genotype and immunophenotype which may be guiding the classifications of primary immunodeficiency diseases. For clinicians, identifying a disease-causing variant in the monogenic form of CVID could assist with the implementation of specific therapeutic options, such as hematopoietic stem cell transplantation or monoclonal antibodies in some CVID-like disorders, as well as implicating the need to elaborate innovative treatment modalities, such as gene therapy, in the future [26,27,43].

Elucidating the role of epigenetic alterations in the immunopathogenesis of CVID would be important not only for scientific purposes, but also for the future developments of patient-centered diagnostic, prophylactic, and therapeutic measures. In these children affected with CVID, in whom conventional genetic testing does not provide informative results, miRNAs could serve as a diagnostic biomarker-oriented approach. To address the miRNA investigations in the clinic, determining the existence of potential epigenetic susceptibility to given exposures might also contribute to a novel prophylactic approach. It could facilitate avoiding exposures to environmental factors that might ultimately trigger the development of B cell disorders resulting in a failure in antibody production and development of CVID. Early-life prophylactic interventions in the context of environmental, nutritional, or immunization measures could be implemented in children in high-risk families when other close family members are affected by CVID. The prophylactic measures also apply to these children, in whom the diagnosis of CVID has already been established to minimize the risk of exacerbation of the clinical course of CVID with infectious, autoinflammatory, autoimmune, or lymphoproliferative disorders [94].

The future perspective of the miRNA approach is pharmacoepigenetics, a novel therapeutic modality serving to improve the health and quality of life of children affected with CVID. The miRNA-based therapeutic strategies might include alterations of miRNA expression specific to CVID pathogenesis and implicate developing targeted treatment with miRNA modifications that could potentially alleviate disease severity or have a curative effect. The epigenetic approach might pave the way to posttranscriptional and posttranslational modifications of pharmacokinetics and pharmacodynamics of various drugs used in the treatment of CVID to improve effectiveness. The assessment of differential miRNA expression in response to immunoglobulin replacement therapy (Ig-RT) might open a new perspective and warrant the need to assess the role of miRNAs as biomarkers of immunopathology in CVID and precision medicine interventions [73]. Moreover, the administration of drugs used in the treatment of CVID could serve as epigenetic modifiers of miRNA effects on specific gene expression [95].

Unraveling the molecular genetic and epigenetic background of CVID might be helpful in developing personalized approaches to treatment, monitoring, and long-term care of affected patients [96]. Immunogenetics paves the way for the development of personalized precision medicine and implementation of modern curative therapies, such as gene therapy or the use of biological molecules, and also assists decisions on the use of immunoglobulin replacement therapy and hematopoietic cell transplantation [97,98,99].

## 7. Conclusions and Future Direction

In a vast majority of children affected with CVID and their relatives, Mendelian patterns of inheritance cannot be applied, reflecting the heterogeneity of clinical manifestation and the complexity of phenotype-genotype correlations. Beyond the monogenic etiology of CVID, other genetic causes include digenic or polygenic background, somatic variants, structural variations, and epigenetic alterations. From the initial era of clinical and laboratory diagnosis of pediatric CVID, the present clinical immunology enters the future era of molecular diagnosis and integrative approaches recapitulating findings in genomics, functional validations, epigenetic profiling, proteomics, and metabolomics. These advanced molecular studies exploring complex multiomics in conjunction with the genetic background of CVID will help to comprehend its composite pathogenesis to aim at patient-tailored precision medicine. The key initiative for the future is improving the accessibility to immunogenetic studies, further implicating challenges and opportunities for precision therapies and improving pediatric CVID patients’ quality of life.

## Figures and Tables

**Figure 1 ijms-25-09999-f001:**
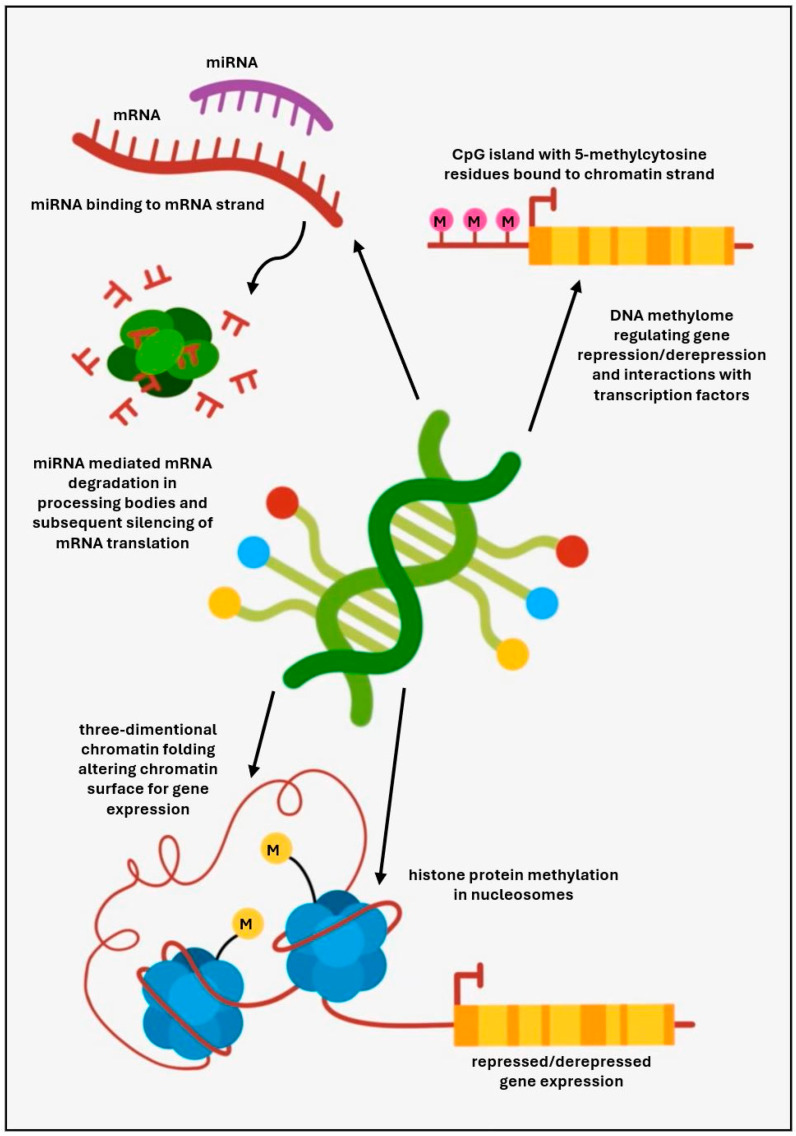
Epigenetic mechanisms regulating gene expression relevant for CVID.

**Table 1 ijms-25-09999-t001:** Categories of immune pathways and a spectrum of variants in genes associated with monogenic CVID according to Peng et al. [28].

**Predominantly Antibody Deficiencies Associated with CVID Phenotypes**					
**BCR Costimulatory B Cell Surface Proteins**	**TNF Superfamily Receptors and Ligands**	**Lipid Signaling Molecules**	**Actin Cytoskeleton Regulators**	**Transcription Factors Mediating Differentiation and Crosstalk**	**Metabolic Mitochondrial and Glycosylation Pathways**
*CD19* *MS4A1/CD20* *CR2/CD21* *CD81*	*TNFSF13B*/BAFF/BLYS/TALL1 *TNFSF13*/APRIL *TNFSF12*/TWEAK *TNFRSF13C*/BAFF-R *TNFRSF13B*/TACI *TNFRSF17*/BCMA	*PIK3CD* *PIK3R1* *PTEN* *PIK3CG* *TTC7A*	*CXCR4* *RAC2* *ARHGEF1* *TTC7A* *PSTPIP1* *DOCK8* *WAS*	*NFKB1* *NFKB2* *IKZF1*/IKAROS	*MAGT1* *ATP6AP1* *PGM3* *TNRT1* *FNIP1* *SBDS* *TAFAZZIN*
**Hypomorphic variants in other genes associated with predominantly antibody deficiencies**					
*BTK* *TCF3* *FNIP1* *ICOS* *CTNNBL1* *ZRSR2*					
**Genes associated with immune dysregulation disorders**					
**Transcriptional regulators of central and peripheral tolerance**		**Membrane-bond organelle dynamics**		**Genes related to lymphoproliferative conditions**	
*AIRE* *FOXP3* *STAT3* *SOCS1* *BACH2*		*CTLA4* *LRBA* *SEC61A1* *SH3KBP1* *DEF6* *SAMD9*		*CD27* *CD70* *MAGT1* *SH2D1A* *PRKCD* *STXBP2* *UNC13D* *FASLG*	
**Genes in combined cellular and humoral immunodeficiencies**					
**Genes associated with T cell signaling regulators**			**Genes associated with epigenetic regulation**		
*ICOS* *CTLA4* *FOXP3* *GATA2* *RFXANK* *LCK* *IL21R*			*DNMT3B* *ZBTB24* *IGH* *KMT2D/*MLL2 *KDM6A/*UTX		
**Genetic underpinnings of autoinflammatory disorders**					
*DCLRE1C/*Artemis *ADA2* *RNF31* *TNFA1P3* *PLCG2* *NLRC3, NLRC4, NLRP2, NLRP3, NLRP12*					

**Table 2 ijms-25-09999-t002:** Digenic CVID and epistatic effect.

Digenic Variants		Product Functions	Clinical Phenotype		Immunodeficiency	Authors (References)
Genes	Variants		Digenic Proband	Monogenic Variant		
*TNFSFR13B/TACI* (17p11.2)	rs34557412 C104R	T-cell independent CSR, MyD88 pathway	CVID Systemic lupus erythematosus	Mild cytopenia Antibody deficiency	Defective T cell dependent and T cell independent B cell differentiation and activation	Ameratunga et al. [44]
*TCF3* (19p13.3)	T168fsX191	T-cell dependent and independent CSR, AID pathway		Antibody deficiency Arthritis Diabetes mellitus		
*NFKB1* (4q24)	(c.1149delT/p.Gly384Glu*48)	Key cellular driver of inflammation and immunity	CVID Inflammatory bowel disease Thrombocytopenia	Asymptomatic	Th1-polarized T cell population B cell lymphopenia and B cell naïvete	Dieli-Crimi et al. [45]
*NOD2* (16q12.1)	rs 5743272 (p.His352Arg)	Inflammatory response to pathogens NF-κB pathway		Crohn disease		
*LRBA* (4q31.3)	C7885delA (p.R2629fs)	Peripheral B cell tolerance, stimulation of T regulatory cell development and functions, CTLA 4 pathway	Antibody deficiency Recurrent airway infections Sepsis Inflammatory bowel disease Throbocytopenia Autoimmune hemolytic anemia	Immunodeficiency with autoimmunity	Decreased ability of T regulatory cells to control T effector cells Defective peripheral B cell tolerance Immunodeficiency	Massaad et al. [46]
*NEIL3* (4q34.3)	rs200055050 (p.D132V)	Regulation of lymphoid cell proliferation, peripheral B cell tolerance		Asymptomatic High levels of autoantibodies		
*CTLA4* (2q33.2)	rs1581573923 (p.Y139C)	Negative regulator of T cell responses, expressed in activated T cells and T regulatory cells	CVID Recurrent airway infections Lymphoid hyperplasia Gastroenteritis Hypothyroidism	Asymptomatic	Impaired memory B cell and plasma cell development T cell hyperactivity	Sic et al. [47]
*JAK3* (19p13.11)	rs200077579 (p.R840C)	Signal transduction from JAK3-associated cytokine receptor common γ chain		Hashimoto thyroiditis		

## Data Availability

The datasets generated during and/or analyzed during the current study are available from the corresponding author on reasonable request.

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
