# Peer review of "Immunogenetic Landscape in Pediatric Common Variable Immunodeficiency"

_ijms, 2024, doi:10.3390/ijms25189999_

Round 1
Reviewer 1 Report
Comments and Suggestions for Authors
Aleksandra Szczawińska-Popłonyk and colleagues presented a review article titled on ‘Immunogenetic landscape in pediatric common variable immunodeficiency’. The study brought points on some aspects of genetics in common variable immunodeficiency, including the monogenic, digenic, and polygenic models of inheritance exemplified by a scope of genes relevant to Common variable immunodeficiency pathophysiology. I would like to forward the following comments and suggestions:
1/ I didn’t see much more detail why this review article is important and what can add to the scientific literature.
2/ This review paper is heavily overlapping with the paper published in 2022 by the same authors, Link below. What is special in this review?
https://www.ncbi.nlm.nih.gov/pmc/articles/PMC8964589/
3/ The same statements and paragraphs are copied from previous papers without changing meaning.
Author Response
1/ I didn’t see much more detail why this review article is important and what can add to the scientific literature.
The comprehensive review article „Immunogenetic landscape in pediatric common variable immunodeficiency” summarizes the current aspects of the genetic and epigenetic background of CVID. The disease is characterized by diverse clinical symptomatology and complex genetic background. While in recent years, the approach to the molecular diagnosis of CVID has been revolutionized, it is crucial to comprehend the complex immunogenetic pathophysiology of this disease. This article provides important data on the complexity of CVID for specialists in different fields of medicine to improve the molecular diagnosis of CVID and ultimately, to improve patients’ access to modern therapeutic procedures.
2/ This review paper is heavily overlapping with the paper published in 2022 by the same authors, Link below. What is special in this review?
https://www.ncbi.nlm.nih.gov/pmc/articles/PMC8964589/
The scope of the review published in the European Journal of Pediatrics by Szczawińska-Popłonyk et al. is an analysis of clinical symptomatology, infectious complications, immune dysregulation disorders, and therapeutic modalities. Genetic aspects of CVID are a minor part of this article. Certainly, some aspects of the CVID molecular pathophysiology are similar, but the present review is much broader with a much more discerning focus on immunogenetics.
3/ The same statements and paragraphs are copied from previous papers without changing meaning.
To avoid overlapping paragraphs in both articles, I have changed the introduction section. However, as a specialist in pediatrics and clinical immunology, I would like to preserve and not change the meaning of some remarks important for clinicians, such as a few sentences on developing antibody response in young children making the diagnosis of pediatric CVID challenging.
The revised Introduction section is displayed in green.
Reviewer 2 Report
Comments and Suggestions for Authors
The manuscript is well written, and there are only minor points that need to be addressed in the revised version:
In general, the list of references should be improved
1. When talking about PAD, the last update of IUIS classification should be included
Tangye SG, Al-Herz W, Bousfiha A, Cunningham-Rundles C, Franco JL, Holland SM, Klein C, Morio T, Oksenhendler E, Picard C, Puel A, Puck J, Seppänen MRJ, Somech R, Su HC, Sullivan KE, Torgerson TR, Meyts I. Human Inborn Errors of Immunity: 2022 Update on the Classification from the International Union of Immunological Societies Expert Committee. J Clin Immunol. 2022 Oct;42(7):1473-1507.
2. A recent and important paper that summarizes genetic B-cell defects:
Tangye SG, Nguyen T, Deenick EK, Bryant VL, Ma CS. Inborn errors of human B cell development, differentiation, and function. J Exp Med. 2023 Jul 3;220(7):e20221105.
3. Please try to persistently use the same abbreviations and uniform terms for immune cells description (f.e. CD4 T cell needs + in superscript).. When using the term “CD4 T helper cell” it should be more precisely defined by adding specific markers that define the cell population.
4. Please, separate the references and use them at the end of each sentence instead of putting more of them at the end of the paragraph – corrections should be made throughout the text.
So, please add the reference at the specified positions:
1st page Line 20, 2nd paragraph (after innate immunity)
1st page Line 5 3rd P (after signaling molecules)
Please separate reff 11, 19, and 23 and use them after citing them within paragraph 2 (2nd P)
Add the reff in the 4th Paragraph, 4th page, 2nd paragraph 5th page, last paragraph 6th page, 1st paragraph 9th page, 1st paragraph of the 9th page, the end of the 2nd paragraph of the 11th page
5. The conclusion and future direction part should be made by the authors themselves, so you should not cite other authors and papers there (ref 91-99). Please try to reduce this part by providing short and clear take-home messages
Comments on the Quality of English LanguageN/A
Author Response
In general, the list of references should be improved
- When talking about PAD, the last update of IUIS classification should be included
Tangye SG, Al-Herz W, Bousfiha A, Cunningham-Rundles C, Franco JL, Holland SM, Klein C, Morio T, Oksenhendler E, Picard C, Puel A, Puck J, Seppänen MRJ, Somech R, Su HC, Sullivan KE, Torgerson TR, Meyts I. Human Inborn Errors of Immunity: 2022 Update on the Classification from the International Union of Immunological Societies Expert Committee. J Clin Immunol. 2022 Oct;42(7):1473-1507.
- A recent and important paper that summarizes genetic B-cell defects:
Tangye SG, Nguyen T, Deenick EK, Bryant VL, Ma CS. Inborn errors of human B cell development, differentiation, and function. J Exp Med. 2023 Jul 3;220(7):e20221105.
According to the suggestions, both important articles have been added to the reference list to improve the quality of the review
- Please try to persistently use the same abbreviations and uniform terms for immune cells description (f.e. CD4 T cell needs + in superscript).. When using the term “CD4 T helper cell” it should be more precisely defined by adding specific markers that define the cell population.
I made corrections in the description of T cell populations, adding CD4+ and CD8+, and added a CXCR5+ T helper CD3+CD4+CD45RO+CD185+ immunophenotype to describe the subset more precisely
- Please, separate the references and use them at the end of each sentence instead of putting more of them at the end of the paragraph – corrections should be made throughout the text.
So, please add the reference at the specified positions: 1st page Line 20, 2nd paragraph (after innate immunity), 1st page Line 5 3rd P (after signaling molecules)
Please separate reff 11, 19, and 23 and use them after citing them within paragraph 2 (2nd P)
Add the reff in the 4th Paragraph, 4th page, 2nd paragraph 5th page, last paragraph 6th page, 1st paragraph 9th page, 1st paragraph of the 9th page, the end of the 2nd paragraph of the 11th page
I revised the references throughout the text thoroughly and separated them adding them at the end of sentences. Please note, that at some points I had to preserve citing more than one paper (eg. ref 85,86,87) as they concern the same data.
- The conclusion and future direction part should be made by the authors themselves, so you should not cite other authors and papers there (ref 91-99). Please try to reduce this part by providing short and clear take-home messages
The conclusion and future directions section has been rewritten and reduced. It provides important remarks on CVID immunogenetics and the future of integrated genomic approaches. All references have been removed from this section
All important corrections in the manuscript have been highlighted in red.
Round 2
Reviewer 1 Report
Comments and Suggestions for Authors
There is some improvement from the previous version but the only change they made is the introduction part and part of conclusion. I recommend the paper to be rewritten or clearly justify in detail why this review is unique from the previously published paper before considering to submit. As this review is a modification of their previous work, I didn't see a significant contribution to the field.
Round 2
Please, find below my explanations about the originality of the manuscript and its contribution to the field of CVID immunogenetics
- The manuscript about pediatric CVID published in Eur J Pediatr discussed different aspects of CVID in children than the current article. It reviewed the burden of CVID in children, clinical phenotype with infectious and non-infectious complications, the pathophysiology of immune deficiency as well as therapeutic modalities. The immunogenetics section was only a minor part of it and a limited number of genes were described. Therefore, the contents of the previous and the current manuscripts are different and the goals of these two works are diverse.
- Other (original) manuscripts on genetics in CVID describe novel genes or their distribution in selected populations. However, please note, that they focus on the monogenic background of the disease. Therefore, this article is a comprehensive review, expanding the discussion on molecular genetics in CVID by adding different models of immunogenetics.
- In the literature, numerous articles describe the results of genetic testing and distribution of variants in the monogenic form of CVID among patients, with different ethnicity of cohorts (eg. American, Swedish, Iranian, selected European, etc.). In contrast to these articles, in the current work, our point of view is not limited to the monogenic CVID, but we also discuss broader the immunogenetic phenomena. We also classify genes relevant to CVID according to their pathophysiological pathways.
- The manuscript is a review aimed at discussing the immunogenetics of CVID, gathering, and highlighting data scattered in the literature, and citing them appropriately. This is not an original work aimed at breakthrough experiments and discoveries and therefore, this manuscript cites a couple of the same (as in the previous article) positions in the CVID literature, eg. works by Seidel et al., Odnoletkova et al., Bogaert et al., or Christiansen et al., and also my work.
Round 3
Reviewer 1 Report
Comments and Suggestions for Authors
Revised version of the manuscript has been significantly improved. No issues identified at this stage.